# Artificial Intelligence in Chest Radiography—A Comparative Review of Human and Veterinary Medicine

**DOI:** 10.3390/vetsci12050404

**Published:** 2025-04-25

**Authors:** Andrea Rubini, Roberto Di Via, Vito Paolo Pastore, Francesca Del Signore, Martina Rosto, Andrea De Bonis, Francesca Odone, Massimo Vignoli

**Affiliations:** 1Department of Veterinary Medicine, University of Teramo, Piano d’Accio, 64100 Teramo, Italy; mrosto@unite.it (M.R.); adebonis@unite.it (A.D.B.); mvignoli@unite.it (M.V.); 2MaLGa, DIBRIS, University of Genoa, Via Dodecaneso 35, 16146 Genoa, Italy; roberto.divia@edu.unige.it (R.D.V.); vito.paolo.pastore@unige.it (V.P.P.); francesca.odone@unige.it (F.O.)

**Keywords:** artificial intelligence, deep learning, chest radiography

## Abstract

Artificial intelligence (AI) could enhance the field of radiology in both human and veterinary medicine by making diagnoses faster and more accurate. In human healthcare, AI assists in detecting diseases such as pneumonia and COVID-19, supporting physicians in pattern recognition and outcome prediction. However, human oversight remains essential due to data limitations and ethical concerns. In veterinary medicine, the use of AI is still limited due to several factors, including the lack of large databases, anatomical differences between animal breeds, and limited research in this field. Focusing on species with less anatomical variability, such as cats, and encouraging interdisciplinary collaboration could foster its development. Despite its potential, the radiologist’s expertise remains crucial. In this context, AI can be seen as a valuable support tool in the daily practice of radiology.

## 1. Introduction

The term “artificial intelligence” (AI) refers to the use of computer systems designed to solve specific problems by simulating human reasoning [1]. A key trait of AI is its capacity to adjust solutions dynamically in response to evolving circumstances, mirroring human adaptability. While AI is designed to mimic specific cognitive functions, it often surpasses human capabilities in tasks such as processing vast amounts of data, recognizing complex patterns, and performing high-speed analyses [2]. AI operates through computational models and algorithms: structured sets of mathematical rules and coded instructions that transform input data into meaningful outputs, enabling problem-solving across various domains. Machine learning (ML) is a core AI methodology that enables computers to learn from examples rather than through explicit programming. By analyzing large datasets, these systems can identify patterns, generate predictions, and support decision-making processes [3]. Building upon these foundations, deep learning (DL) represents a significant advancement in AI research and now forms the basis for most recent AI innovations in the field. This methodology employs Artificial Neural Networks (ANNs) inspired by biological neural systems, processing information through a hierarchical structure comprising an input layer, multiple hidden layers, and an output layer. Through its distinctive incorporation of numerous hidden processing layers, deep learning efficiently analyzes complex, multidimensional veterinary data that traditional methods struggle to interpret. These capabilities have enabled sophisticated applications across veterinary medicine, including advanced diagnostic imaging analysis and complex clinical parameter interpretation [4]. AI has introduced groundbreaking improvements in the interpretation of X-rays, ultrasound images, magnetic resonance imaging (MRI), and computed tomography (CT) scans. Its impact extends beyond mere automation, fundamentally reshaping diagnostic processes to enhance speed, accuracy, and efficiency [5]. One of the main advantages of AI in imaging is its ability to significantly speed up analysis. Traditional methods, which are often slow and prone to human error, take a long time [6]. AI, on the other hand, can process and examine medical images in extremely short times, offering rapid diagnoses, a crucial aspect especially in emergency situations, where every second counts [7,8].

In addition to speed, AI also improves diagnostic accuracy. By analyzing vast medical datasets, algorithms can recognize patterns and abnormalities that might escape the human eye. This increased accuracy makes it possible to reduce diagnostic errors and ensure that patients receive appropriate and timely treatments [7].

In human medicine, the radiologists’ error rate in interpreting diagnostic images has been thoroughly analyzed, considering the potential consequences for both patients and healthcare facilities [9,10]. Errors in the interpretation of diagnostic images can arise from various factors, such as missed lesions located in anatomically overlooked areas (commonly referred to as ‘blind spots’), suboptimal use of imaging settings, overlapping pathological structures, or atypical presentations of certain diseases. These are common contributors that can compromise diagnostic accuracy [10]. Despite significant advancements in professional expertise, clinical knowledge, and technological innovation, the error rate in this field has remained surprisingly consistent over time, for example, staying below 15% in chest radiographic studies [10].

For this reason, several studies in human medicine suggest the use of AI-based softwares as support for the radiologist in image acquisition and interpretation [10,11]. However, in veterinary medicine, the error rate that characterizes the analysis of radiographic images has not been thoroughly studied [12,13]. Thus far, 18 studies have been published on the use of AI applied to the interpretation of chest radiographs (CXR) of dogs and cats. Of these, about six articles have focused on the detection of major alterations in cardiac silhouette. For example, in the study by Li and colleagues [14], they developed a convolutional neural network (CNN) based on the Visual Geometry Group 16 model to detect left atrium enlargement in lateral chest radiographs. The study used a database consisting of 792 radiographic images, which were then classified as “positive” or “negative” about left atrium enlargement. The images were subjected to classification by both a CNN and certified radiologists with the model achieved an overall accuracy of 82.71%, with a sensitivity of 68.42% and a specificity of 87.09%.

In research conducted by Fitzke [15], approximately 2.5 million thoracic and extra-thoracic radiographs of dogs and cats were used to develop and train a DL model aimed at identifying various abnormalities. The results obtained proved to be quite encouraging, showing a low false positive rate (ranging from 0 to 0.057) and high sensitivity (ranging from 0 to 0.962). AI also presents significant limits that can be significant for its use in the daily routine. ML and DL models can be inaccurate due to issues with training data, such as insufficient or unrepresentative datasets, leading to false positives and negatives. Furthermore, the use of ML models on different patient populations from those used during the training phase may generate bias, considering different variables such as age, breed, morphology, differences in imaging techniques, and labeling modalities [16]. If such biased or inaccurate models are applied on a large scale, they could negatively affect the quality of care provided to many veterinary patients and mislead the vet.

A key issue is that AI algorithms are primarily trained and validated using large human medicine databases, whereas veterinary datasets are generally more limited [17].

Another ethical problem is that trust in artificial intelligence can decline, particularly with deep learning systems like CNNs, which, despite their accuracy, makes it difficult to understand what is behind AI decisions, such as the classification of radiological anomalies [18].

Beyond interpretability, ethical concerns also arise regarding patient data privacy in radiological AI models [17] and the environmental impact of energy-intensive AI systems [19].

Initially, we filtered the selected articles based on the anatomical domain, including only AI models that focused on CXR in both human and veterinary fields.

Then, this review adopted a critical and interpretative approach to analyze the most recent and cited applications of AI in the field of radiology, with a specific focus on comparing its use in the interpretation of CXR in human and veterinary medicine. Rather than strictly following a systematic review model, our aim was to offer a reasoned overview of currently available AI-based tools by selecting recent and clinically relevant studies in the context of diagnostics and patient care. The objective is to explore the similarities and differences between the two fields, both in terms of methodological approaches and performance, through a critical evaluation of the most recent literature. Particular attention was given to studies that not only present technical innovations but also offer practical insights into their applicability in real-world clinical settings. By identifying key trends, potential discrepancies in outcomes, and sector-specific challenges, this review aims to highlight the main differences among the latest publications in human and veterinary medicine. Ultimately, we seek to promote a collaborative and interdisciplinary understanding that can enhance the development of future studies in the field of AI, particularly within veterinary medicine.

## 2. Applications of Artificial Intelligence in Human Medicine: Chest Radiographs

Several recent studies have analyzed the performance of radiologists in interpreting diagnostic images, comparing the results obtained with and without AI assistance. For example, in the case of CXRs, the use of AI has been shown to significantly improve the identification of abnormalities such as active tuberculosis [20], malignant nodules [21], and other pathologies of the greater thoracic district [22].

In this context, the AI system generates probabilities associated with specific diseases, sometimes accompanied by indications of the locations of abnormalities. Clinicians can supervise and integrate this information during or after image interpretation. A relevant example is the DL algorithm developed by Nam and colleagues [23], designed to detect 10 common abnormalities in CXR images, including pneumothorax, mediastinal thickening, pneumoperitoneum, nodules/masses, consolidation, pleural effusion, atelectasis, fibrosis, calcifications, and cardiomegaly. This algorithm also offers visual localization of abnormalities, providing additional support to physicians in the diagnostic process.

In the latter study [23], DLAD-10 (deep learning algorithm detecting) was trained with 146,717 radiographs of 108,053 patients using a ResNet34-based neural network with lesion-specific channels. As a yardstick, the performance of DLAD-10 was compared with a same-day CT-confirmed dataset (normal: abnormal) and an open-source dataset (PadChest; normal: abnormal) and accordingly compared with that of three radiologists. DLAD-10 correctly classified significantly more critical abnormalities (95.0% (57 out of 60)) than the pool of radiologists (84.4% (152 out of 180)). Ultimately, this DL algorithm showed excellent performance, improving radiologists’ skills and reducing the time to report critical and urgent cases.

Similarly, Seah and colleagues in 2021 [24] conducted a study in which twenty radiologists reviewed CXR results of 127 clinical cases with and without the assistance of a DL algorithm and found that radiologists assisted by the DL algorithm showed much better reading performance, with higher areas under the curve (AUC) when assisted by the DL algorithm (AUC, 0.808; 95% confidence interval (CI)) than when not assisted (AUC, 0.713; 95% CI). The DL algorithm significantly improved radiologists’ classification accuracy for 102 (80%) of the 127 clinical findings and was statistically non-inferior for 19 (15%) cases; moreover, none of these radiographic studies showed a decrease in accuracy when radiologists used the DL algorithms.

Furthermore, a more recent study by Banerjee et al. (2025) [25] reinforced the role of AI in radiology by demonstrating its effectiveness in cancer detection, particularly for chest radiographs. The study highlights AI’s potential to assist radiologists in identifying lung nodules and other malignancies, which aligns with previous findings on the benefits of deep learning-assisted diagnoses [25].

However, while AI has demonstrated great promise, a recent study by Bottazzi, Ferrante, and Cheplygina (2024) [26] emphasized that the dataset used to train AI models significantly impacts their robustness and generalizability. Their study on the NIH CXR14 dataset highlights how different sources of data annotation affect model performance, raising important concerns regarding dataset bias in AI applications for chest radiographs.

Carloni and Tsaftaris (2025) [27] introduced a framework to enhance AI model robustness against domain shifts in chest radiographs. Their study demonstrated how domain variability significantly affects diagnostic reliability, reinforcing the need for domain-adaptive AI.

Similarly, Liao and Xia (2025) [28] addressed label noise in CXR datasets by improving diagnostic accuracy despite noisy annotations. This research is particularly relevant to ensuring AI models remain reliable, even with imperfect training labels.

Additionally, Pedrosa et al. (2024) [29] presented an Anatomically-Guided Inpainting technique to reconstruct missing lung regions in CXR images. This approach allows AI models to handle incomplete scans effectively, making it particularly useful for real-world clinical applications.

The added value of AI assistance is particularly evident in specific situations such as emergencies. CXR is a simple and widely accessible imaging modality; however, its interpretation is not easy and often requires a high level of expertise and experience. Many studies have found substantial discrepancies in the interpretation of CXR images in the emergency department, ranging from 0.3% to 17% [30,31]. This type of discordant misinterpretation of critical cases could directly influence the clinical course and outcomes of patients. In addition, emergency room physicians often have little time or opportunity to reach an on-call radiologist for consultations [32]. Hwang [22] studied whether the application of a commercially available DL algorithm could improve physicians’ reading performance for clinically relevant abnormalities on CXR scans in the context of emergency management. The assistance of the DL algorithm improved the sensitivity of radiology physicians’ interpretation from 65.6% to 73.4%. Subsequently, in 2020, Kim and colleagues [33] reported that with DL algorithm support, physicians’ diagnostic performance for pneumonia improved (sensitivity: 53.2% to 82.2%; specificity: 88.7% to 98.1%).

The study of Harmon [34] showed that a DL algorithm can achieve 90.8% accuracy with 84% sensitivity and 93% specificity in detecting COVID-19 pneumonia on CT scans. In addition, further studies have suggested that AI can support radiologists in distinguishing COVID-19 from other pulmonary infections on both CXR [20] and CT scans of the chest [34].

AI-based models have the potential to integrate multimodal data collected from patients, thus transforming the process of detection, diagnosis, and triage of suspected COVID-19 cases. In particular, the study by Ippolito [35] showed that AI can be extremely useful in daily clinical practice, especially in emergency departments where the large number of patients and the need for rapid responses can significantly affect diagnosis.

AI makes it possible to identify and distinguish different patterns of lung infections, improving clinical decision making, reducing response time, and increasing diagnostic confidence. As highlighted, AI systems can also detect characteristic signs of bacterial pneumonia, representing a crucial resource in settings where access to CT is limited or diagnostic expertise is insufficient.

The study also showed that AI can classify patients into three main categories: COVID-19 positive, pneumonia positive and healthy subjects, ensuring high accuracy and low error rates. These results help to strengthen the diagnostic confidence of radiologists, providing an effective tool to improve clinical care [35].

Particularly interesting is the 2024 study by Obuchowicz and colleagues [36] in which a novel radiographic motion simulation network was introduced that integrates U-Net with LSTM networks to simulate and predict respiratory motion of the lungs from single-phase chest radiographs. Then, a spatial transform network is applied for precise image deformation to reflect the real respiratory motion. The performance of the network is evaluated by qualitative and quantitative methods. This approach improves diagnostic capabilities by providing information on lung dynamics from static radiographs, offers a noninvasive alternative for lung function assessment, and increases diagnostic efficiency by extracting detailed information from routine chest radiographs.

A prospective, randomized, controlled study by Nam [37] involved 10,476 participants undergoing CXR during health checkups. Participants were randomly assigned to two groups: one using an AI-based system and one without AI. The primary objective was to measure the detection rate of treatable lung nodules calculated as the ratio of positively identified radiographs to total radiographs. Secondary objectives included rates of false reports, positive reports, and detection of malignant nodules and lung cancer. The detection rate of treatable lung nodules was significantly higher in the AI group (0.59% vs. 0.25%). The study highlights the potential of AI in improving the detection of pulmonary nodules in CXR scans during health screenings but emphasizes the need for further multicenter research to confirm these findings and evaluate the clinical impact on a large scale [37]. However, a recent article was published by Garza-Frias [38] in which the use of an AI software-based (Qure.AI, Version 3.1.6) index to assess CXR images and predict the development of heart failure within one year of examination was investigated. A multicenter retrospective study of 1117 patients (mean age 67.6 years) with no pre-existing diagnoses of heart failure was conducted. Of these, 413 developed heart failure within one year of examination, while 704 did not. CXR images were analyzed with the qXR-HF (Qure.AI, Version 3.1.6) model, which provided information on cardiac silhouette, pleural effusion, and lung pattern. The results showed an AUC of 0.798 (95% CI 0.77–0.82), accuracy of 73%, sensitivity of 81%, and specificity of 68%. These data demonstrate the potential of opportunistic screening using AI in radiology, highlighting how automated analysis of CXR results can proactively identify patients at risk of developing heart failure, enabling timely and targeted interventions [38].

To address these issues, Park and Kooi (2024) [4] proposed Positive-Sum Fairness, an AI training methodology that ensures equitable diagnostic performance across demographic groups without degrading accuracy for any subgroup.

Furthermore, Queiroz et al. (2024) [39] introduced Backbone Foundation Models, which enable fairness evaluation in AI-based chest radiography without needing explicit demographic data. Their study suggests that AI fairness can be improved without compromising diagnostic accuracy.

## 3. Applications of Artificial Intelligence in Veterinary Medicine: Chest Radiographs

In human medicine, the use of AI is now a well-established reality, supported by a robust body of scientific literature. In contrast, in veterinary medicine, the exploration and publication of AI applications are still in their early stages. Nevertheless, some studies have begun investigating potential uses of AI in specialized fields, such as veterinary radiology.

In a study conducted by Yoon and colleagues [40], algorithms were employed to automatically analyze CXR images with the aim of detecting abnormalities in the cardiac silhouette, lung parenchyma, mediastinum, and pleural space, and distinguishing them from normal images. Initially, the study used models based on the Bag of Features (BOFs) technique, but the most promising results were achieved using CNNs. The CNNs reached accuracy levels ranging from 92.9% to 96.9% and sensitivities from 92.1% to 100%. In comparison, BOF models performed less effectively, with accuracy ranging between 79.6% and 96.9% and sensitivity between 74.1% and 94.8%. These findings highlight the superior performance of CNNs over BOF-based models.

Another interesting application of AI in veterinary radiology is found in the study by Kim and colleagues [41], who used commercial software (Vetology Innovations, San Diego, CA, USA) to detect cardiogenic pulmonary edema in 500 canine chest radiographs. Nineteen images were excluded due to technical issues or poor quality. Among the evaluated images, the system achieved an accuracy of 92.3%, with a specificity of 92.4% and a sensitivity of 91.3%. Notably, the negative predictive value (NPV) was 99%, indicating high reliability in confirming the absence of disease. However, the positive predictive value (PPV) was only 56%, suggesting that a positive result still requires a confirmation. It is also important to note that the number of radiographs used to train the system was significantly lower than the data volumes typically used in human medicine studies, and this could introduce bias.

The same software was later employed in two additional research studies. The first, conducted by Müller and colleagues [42], tested the software’s ability to detect pleural effusion in radiographs from 62 dogs. The results showed an accuracy of 88.7%, sensitivity of 90.2%, and specificity of 85.7%. However, it is worth mentioning that diagnoses were based solely on radiologist interpretation, without clinical, laboratory, or advanced imaging confirmation. This represents a limitation, as noted by the authors, considering that radiologists’ accuracy in detecting pleural effusion can vary significantly, ranging from 67% to 92% [43]. As highlighted in the study, further research is needed to evaluate the true accuracy of this AI system [42].

An innovative approach was taken in the study by Pomerantz and colleagues [44] where the same AI software (Vetology Innovations, v. veterinary teleradiology) was used to detect pulmonary nodules and masses in 56 canine patients. The software’s results were subsequently compared with computed tomography (CT) images of the same patients, which is considered the gold standard for detecting pulmonary nodules and masses [45]. The system achieved an accuracy of 69.3%, with a sensitivity of 55.4% and a specificity of 93.7%. While these performance metrics are lower than those seen in previous studies, they nonetheless confirm the potential of AI as a clinical support tool and diagnostic support. At the same time, they underscore the importance of combining AI with radiologist expertise. The decrease in performance compared to Müller’s study [42] may be due to the use of CT as an objective diagnostic reference. Once again, the primary limitation was the small dataset used [44].

Currently, AI applications for interpreting feline chest radiographs remain limited, with only a few studies available, such as those by Banzato [46] and Dumortier [47]. Banzato’s study involved training two deep neural network architectures, ResNet 50 and Inception V3, to recognize common thoracic radiographic findings in cats. These included bronchial patterns, pleural effusion, pulmonary masses, alveolar patterns, pneumothorax, cardiomegaly, and normal findings. The models showed good performance for most diagnostic categories, achieving area under the curve (AUC) values greater than 0.8. However, accuracy was lower for detecting cardiomegaly (AUC > 0.7) and particularly for pulmonary masses (AUC > 0.5).

In the second study, published by Dumortier and colleagues [47] in 2022, the ResNet50V2 neural network was used to classify feline thoracic images. A manual segmentation system was employed to define regions of interest. Although the results were promising, the method’s effectiveness was limited by the small dataset (500 radiographs) and the requirement for human input during segmentation, both of which hinder its practical application in clinical settings.

As for the automatic analysis of the cardiac silhouette, veterinary literature is still quite sparse. One of the most significant studies was conducted by Burti and colleagues [48] who assessed the accuracy of four different CNN models in classifying the presence or absence of cardiomegaly using the Vertebral Heart Score (VHS) as a reference, taking breed variability into account. The best-performing model was based on the ResNet-101 architecture, achieving an AUC of 0.97. The study used a large dataset of 1465 lateral thoracic radiographs, the standard projection for VHS measurement, and concluded that this technology could serve as a valuable support tool for radiologists in clinical practice, especially given the somewhat subjective nature of this type of measurement.

In another study, researchers developed a DenseNet-121 neural network capable of automatically measuring VHS on lateral thoracic radiographs. Although the number of images used was limited (60 radiographs from dogs and cats), the system’s measurements showed high concordance (>0.9) with those of two expert radiologists [49]. Again, a major limitation is that VHS measurements can vary depending on the operator. A similar investigation was conducted by Zhang and colleagues in 2021 [50], in which CNNs were trained to identify anatomical landmarks needed for VHS calculation. The system achieved an average accuracy of 91%, suggesting a promising clinical application to make VHS assessment more objective and less susceptible to inter-operator variability, a common issue that can reduce the effectiveness of such measurements.

## 4. Discussion

The integration of AI into thoracic radiography, both in human and veterinary settings, represents a promising innovation that can help improve diagnostic quality and significantly reduce image interpretation time. Table 1 provides a summary of the main articles discussed in this review. However, it is critical to emphasize that AI systems should not be considered a substitute for the physician or radiologist. Although AI can provide valuable support, offering a “second opinion” and increasing diagnostic confidence, it remains susceptible to errors that only human clinical judgment can identify and contextualize. On the other hand, the study by Rudnay and Kovac (2024) [51] shows how the human factor, both psychological and physical, can lead to errors in interpretation, especially in the field of imaging.

Since the human factor cannot be eliminated, neither in medicine nor in forensic practice, it is essential to recognize its limitations, and for this reason the integration of AI can provide valuable support [51].

The application of AI in medical imaging diagnostics could have negative effects on healthcare. For instance, a machine learning model designed to predict the risk of human pneumonia mistakenly assigned lower risk scores to asthma patients due to biases in the clinical data used for training. This error could potentially result in serious harm to patients [52].

The performance of AI systems hinges critically on their training foundation. Optimal results require large, well-structured datasets that precisely align with the intended diagnostic objectives—a significant challenge in veterinary medicine where such comprehensive data resources remain limited compared to human medical applications. Ensuring data consistency and representativeness can help mitigate biases and improve the reliability of AI-driven medical assessments.

In particular, the application of AI in veterinary medicine is still in its infancy and requires further study to fully assess its potential and adapt its techniques to the specific characteristics of species such as dogs and cats. An additional factor of complexity in veterinary medicine is the variability among different breeds, particularly in dogs, which can affect the accuracy of AI systems and lead to higher error rates. Therefore, it is essential that future research delves into these issues, encouraging increasingly effective and safe integration of these technologies into diagnostic practices, like what is being attempted in human medicine. One possible solution to overcome this limitation could be to begin studies focused on animal species with less phenotypic variability among breeds, such as cats. In this context, the relative homogeneity among feline breeds could facilitate the training of AI systems, and in this sense, it is comparable to the human case, reducing the complexity associated with analyzing more heterogeneous samples. Such an approach could represent a significant step toward the development of more accurate and generalizable AI models in veterinary medicine, opening new perspectives for the application of technology in the field. In this regard, in Figure 1 we depict potential AI tasks with a broad adoption in chest radiography analysis for human settings that could be useful to extend in the veterinary domain, such as heart segmentation and cardiac silhouette feature extraction through anatomical landmark detection.

Recent advances in landmark detection techniques for x-ray images have shown promising results that could be valuable for both human and veterinary applications. Di Via et al. [53] conducted a systematic study analyzing whether small-scale in-domain datasets provide any benefit for landmark detection over models pre-trained on large natural image datasets only. Their findings suggest that pre-training with ImageNet may be as effective as in-domain pre-training for anatomical landmark detection in x-ray images, which could simplify implementation in veterinary settings where large in-domain datasets are scarce. Furthermore, in a subsequent study, Di Via et al. [54] proposed a novel self-supervised pre-training approach using diffusion models for few-shot landmark detection in x-ray images, demonstrating good performance with as few as 50 annotated training images. These methodologies could be particularly valuable in veterinary medicine where annotated datasets are limited, especially for automated cardiac silhouette assessment as shown in Figure 1.

In this context, the guidance offered by studies conducted in human medicine is a valuable resource for advancing the veterinary field as well, providing models for research and application of artificial intelligence and other diagnostic technologies. However, as highlighted in this review, veterinary practice makes greater use of cardiac silhouette assessment in CXR images than in the human field, as advanced imaging techniques for cardiac studies are less available. This approach, more widely adopted in the veterinary field, could prove to be an important opportunity for mutual growth and integration between the two fields. By fostering a bidirectional transfer of knowledge, more accurate and personalized diagnostic methods could be developed for both humans and animals. On the other hand, the use of AI is significantly more exploited in the interpretation of CXR scans during emergency situations in human medicine. This could serve as an excellent area to enhance tools in veterinary medicine, given the growing need to manage emergency cases effectively like in the study of Hwang and colleagues [22]. A crucial aspect that differentiates AI studies in human medicine from those in veterinary medicine is the size of the samples used. Studies in human medicine tend to benefit from significantly larger publicly available datasets, allowing for more robust statistical models and more accurate validation of results. In contrast, in veterinary medicine, the smaller sample size can affect the robustness of conclusions, posing a challenge for the effective application of AI. This underscores the importance of promoting cross-sectoral collaborations and developing strategies for augmenting and sharing datasets in the veterinary context as well, to bridge this disparity and maximize the potential of AI in both disciplines. Further studies are necessary to effectively implement the use of artificial intelligence in veterinary radiology, ensuring not only technical advancements but also clinically relevant applications.

## 5. Conclusions

The integration of artificial intelligence into thoracic radiography represents a promising innovation in both human and veterinary medicine, enhancing diagnostic quality and reducing interpretation time. However, AI cannot replace clinical judgment, as it is prone to biases and errors. In veterinary medicine, challenges arise from breed and species variability, as well as the limited availability of structured datasets. Studies on species with lower phenotypic variability, such as cats, could facilitate the training of more accurate models. In fact, in feline thoracic radiography, variations in thoracic conformation, cardiac silhouette appearance in different recumbency, tracheal size, and diaphragm shape are less pronounced among different breeds. Furthermore, promoting data sharing and integration is crucial for developing reliable AI applications in both fields. This review aimed to critically evaluate the current state and future potential of AI applications in thoracic radiography across human and veterinary domains, identifying both shared opportunities and discipline-specific challenges to guide future research and clinical implementation.

## Figures and Tables

**Figure 1 vetsci-12-00404-f001:**
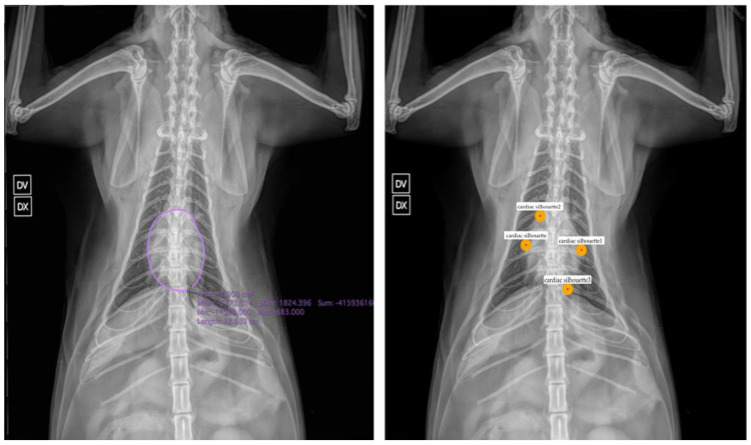
Potential AI tasks we envision on veterinary images, with examples on a cat chest radiograph. Left. Heart segmentation and extraction of metrics. Right. Anatomical landmark detection for analyzing cardiac silhouette. Output images and data from these tasks can potentially reveal information useful for automatic diagnosis in a classification framework.

**Table 1 vetsci-12-00404-t001:** Summary table of the main articles included in this review.

Reference	Task	Species
**Banzato et al., 2021 [46]**	Detecting common radiographic findings	Dog/cat
**Boissady et al., 2021 [49]**	Automatically measuring VHS	Dog/cat
**Burti et al., 2020 [48]**	Classification of cardiomegaly based on VHS value	Dog
**Fitzke et al., 2021 [15]**	Detecting thoracic and extra-thoracic radiographic abnormalities	Dog/cat
**Garza-Frias et al., 2024 [38]**	Early detection of heart failure	Human
**Hwang et al., 2019 [20]**	Identification of tuberculosis, malignant nodules, and other anomalies	Human
**Hwang et al., 2019 [22]**	Use of commercial DL software in emergencies	Human
**Ippolito et al., 2023 [35]**	Distinguish different patterns of lung infections	Human
**Kim et al., 2020 [33]**	Deep learning algorithm for detection of pneumonia	Human
**Kim et al., 2022 [41]**	Presence/absence of cardiogenic pulmonary edema	Dog
**Li et al., 2020 [14]**	Detecting left atrial enlargement	Dog
**Müller et al., 2022 [42]**	Presence of pleural effusion	Dog
**Nam et al., 2021 [23]**	Detection of 10 common abnormalities in CXR scans	Human
**Nam et al., 2023 [21]**	Detection of lung nodules	Human
**Obuchowicz et al., 2024 [36]**	Real-time CXR	Human
**Pomerantz et al., 2023 [44]**	Attendance of pulmonary nodules and masses	Dog
**Seah et al., 2021 [24]**	Effect of a comprehensive deep learning model on the accuracy of CXR interpretation	Human
**Yoon et al., 2018 [40]**	Normal vs. abnormal cardiac silhouette and thoracic portions	Dog
**Zhang et al., 2021 [50]**	Identification of landmarks for calculating VHS	Dog
**Banerjee et al., 2025 [25]**	AI in cancer diagnosis in radiology	Human
**Juodelyte et al., 2024 [26]**	The importance of datasets	Human

## Data Availability

The datasets presented in this article are not readily available due to restrictions from our Institutional Review Board.

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
