# Peer review of "Artificial Intelligence in Chest Radiography—A Comparative Review of Human and Veterinary Medicine"

_vetsci, 2025, doi:10.3390/vetsci12050404_

Round 1

Reviewer 1 Report

Comments and Suggestions for Authors

I reviewed the important review paper entitled Artificial Intelligence in Chest Radiography – A Comparative  Review of Human and Veterinary Medicine .the paper is very important especially as AI is now used in many fields especially veterinary clinic and veterinary aspects. 

The paper need some modifications and revision 

Abstract the COVid must be written at first with a capital letter not abbreviated then you can write after that in an abbreviated form .

Line 22 what about recent advance?

Please in keywords the words must be rewritten again with first capital letter 

Introduction 

How come first references in the intro was  was 17!?

What about references from 1-17 please clarify 

Line 40 how AI operates through com putational models and algorithms? Please justifying this important paragraph with more relevant references for this topic.

What about refrences style how come references number 42 come before number 28 please this paragraph should be managed .

Line 47-55 authors stated that One of the domains most profoundly revolutionized by AI is healthcare please more references are needed in addition this paragraph mist be rewritten again with more details and summarized topic.

Lines 61-63 without no refrences please add two -three for this sentence.

Lines 94_104 about application of AI need more paraphrasing as some errors were present as Grammer errors.

Authors must write the total paragraph about Applications of Artificial Intelligence in the Veterinary Medicine: 219 Chest Radiographs on a simple manner 

Please add more figures with radiography to get the reviewore relevant and clear in idea.

Conclusion must be summarized as small as possible.

Thank you 

I look forward to reviewed the revision versions and I hope to take my recommendation in your consideration.

Comments on the Quality of English Language

The English language should be reorganized with a native English speaker 

Author Response

Comments 1

The paper need some modifications and revision ,Abstract the COVid must be written at first with a capital letter not abbreviated then you can write after that in an abbreviated form . Line 22 what about recent advance? Please in keywords the words must be rewritten again with first capital letter.

Response 1

Thank you for pointing this out. We have fixed the abbreviations in the text. Regarding line 22, it refers to some studies in which AI was successfully used to detect left heart enlargement, pulmonary edema, and pleural effusion in radiographic studies of small animals. I hope the corrections are satisfactory.

Comments 2 How come first references in the intro was  was 17!?

What about references from 1-17 please clarify.

Line 40 how AI operates through com putational models and algorithms? Please justifying this important paragraph with more relevant references for this topic.

What about refrences style how come references number 42 come before number 28 please this paragraph should be managed .

Line 47-55 authors stated that One of the domains most profoundly revolutionized by AI is healthcare please more references are needed in addition this paragraph must be rewritten again with more details and summarized topic.

Lines 61-63 without no refrences please add two -three for this sentence.

Lines 94_104 about application of AI need more paraphrasing as some errors were present as Grammer errors]

Please add more figures with radiography to get the reviewore relevant and clear in idea.

Response 2

Thank you for the revision suggestions. We changed the order of references in the text.

In line 40, we added a brief description regarding the functioning of the main AI models used in the field of imaging diagnostics.

Lines 47-55: We made some modifications to this paragraph.

Lines 61-63: We added some references.

Lines 94-104: We attempted to improve the grammar and syntax.

Regarding the inclusion of additional images, we thought it would be better to use something “of our own,” like the figure in this review, rather than using images taken from other articles. In our initial idea, we had included some images from human medicine articles, but since this is a veterinary medicine journal, we decided to remove them.

Reviewer 2 Report

Comments and Suggestions for Authors

The manuscript provides a thorough exploration of artificial intelligence applications in chest radiography, drawing comparisons between human and veterinary medicine. The discussion is well-structured, engaging with key advancements in AI-driven diagnostics while acknowledging the challenges specific to veterinary applications. However, despite the breadth of the literature cited, the review would benefit from a more critical engagement with existing studies and a deeper exploration of certain key areas.

The current discussion effectively highlights the role of AI in human radiology, particularly its impact on diagnostic accuracy and efficiency. Studies showcasing AI’s performance in detecting pulmonary conditions, such as tuberculosis, pneumonia, and COVID-19, are well-integrated into the narrative. However, the veterinary section does not receive the same depth of analysis. While some key studies are mentioned, the discussion remains largely descriptive rather than critically engaging with the limitations of the cited research. A more detailed examination of the methodologies employed in healthcare AI research (10.1038/s41598-024-76608-2, 10.1109/ISBI53787.2023.10230448), including dataset limitations, model validation, and clinical applicability, would strengthen the discussion. The manuscript would be significantly improved by expanding the comparative analysis beyond general AI performance, focusing on how variations in medical imaging practices between human and veterinary medicine influence AI effectiveness.

The review does well in acknowledging the scarcity of veterinary AI studies, but it does not sufficiently interrogate the implications of this limitation. The issue of sample size in veterinary datasets is mentioned, but the discussion lacks specificity regarding how this limitation affects the generalizability of AI models. A stronger critique of dataset biases and the variability in radiographic techniques across healthcare institutions would be valuable (10.1371/journal.pone.0308758). More references to interdisciplinary efforts that have successfully addressed similar challenges in human medicine could also provide a clearer roadmap for advancing veterinary AI research.

Methodological clarity in literature selection is another area requiring improvement. The review does not explicitly state how studies were chosen for inclusion or whether a systematic approach was used. Without such transparency, it is difficult to assess whether the selection of literature is comprehensive or potentially biased toward studies with more favorable AI outcomes (10.1148/ryai.230502). Clarifying the criteria used for evaluating AI performance across different studies would add rigor to the manuscript.

Another area that deserves further exploration is the ethical dimension of AI-assisted diagnostics. The review briefly acknowledges that AI should serve as a complementary tool rather than a replacement for human expertise. However, the ethical challenges associated with AI in veterinary medicine remain underexplored. Questions regarding liability in cases of AI-assisted misdiagnosis, the need for regulatory frameworks, and the ethical implications of AI-driven decision-making in clinical settings should be discussed more thoroughly. A stronger engagement with veterinary ethics literature could provide valuable insights.

The manuscript is well-written and accessible, but there are areas where conciseness could improve readability. Some explanations of AI fundamentals are somewhat redundant, especially in the introductory sections. While clarity is generally maintained, certain paragraphs would benefit from more direct phrasing to reduce repetition. Additionally, the inclusion of tables summarizing key findings across human and veterinary AI applications would enhance the clarity of comparisons.

Overall, the manuscript presents a relevant and timely discussion on AI in radiographic diagnostics, but there are areas where the argument could be more robust. A deeper engagement with veterinary-specific challenges, a clearer methodological approach to literature selection, and a more critical discussion of ethical considerations would significantly strengthen the review. Revisions addressing these aspects would enhance the manuscript’s contribution to the field.

Author Response

COMMENTS 1

However, the veterinary section does not receive the same depth of analysis. While some key studies are mentioned, the discussion remains largely descriptive rather than critically engaging with the limitations of the cited research. A more detailed examination of the methodologies employed in healthcare AI research (10.1038/s41598-024-76608-2, 10.1109/ISBI53787.2023.10230448), including dataset limitations, model validation, and clinical applicability, would strengthen the discussion. The manuscript would be significantly improved by expanding the comparative analysis beyond general AI performance, focusing on how variations in medical imaging practices between human and veterinary medicine influence AI effectiveness.

The review does well in acknowledging the scarcity of veterinary AI studies, but it does not sufficiently interrogate the implications of this limitation. The issue of sample size in veterinary datasets is mentioned, but the discussion lacks specificity regarding how this limitation affects the generalizability of AI models. A stronger critique of dataset biases and the variability in radiographic techniques across healthcare institutions would be valuable (10.1371/journal.pone.0308758). More references to interdisciplinary efforts that have successfully addressed similar challenges in human medicine could also provide a clearer roadmap for advancing veterinary AI research.

Methodological clarity in literature selection is another area requiring improvement. The review does not explicitly state how studies were chosen for inclusion or whether a systematic approach was used. Without such transparency, it is difficult to assess whether the selection of literature is comprehensive or potentially biased toward studies with more favorable AI outcomes (10.1148/ryai.230502). Clarifying the criteria used for evaluating AI performance across different studies would add rigor to the manuscript.

Another area that deserves further exploration is the ethical dimension of AI-assisted diagnostics. The review briefly acknowledges that AI should serve as a complementary tool rather than a replacement for human expertise. However, the ethical challenges associated with AI in veterinary medicine remain underexplored. Questions regarding liability in cases of AI-assisted misdiagnosis, the need for regulatory frameworks, and the ethical implications of AI-driven decision-making in clinical settings should be discussed more thoroughly. A stronger engagement with veterinary ethics literature could provide valuable insights.

The manuscript is well-written and accessible, but there are areas where conciseness could improve readability. Some explanations of AI fundamentals are somewhat redundant, especially in the introductory sections. While clarity is generally maintained, certain paragraphs would benefit from more direct phrasing to reduce repetition. Additionally, the inclusion of tables summarizing key findings across human and veterinary AI applications would enhance the clarity of comparisons.

Overall, the manuscript presents a relevant and timely discussion on AI in radiographic diagnostics, but there are areas where the argument could be more robust. A deeper engagement with veterinary-specific challenges, a clearer methodological approach to literature selection, and a more critical discussion of ethical considerations would significantly strengthen the review. Revisioni addressing these aspects would enhance the manuscript’s contribution to the field.

RESPONSE 1

Thank you for your suggestions. We have made significant improvements to the section on ethical considerations, including the potential limitations of AI in veterinary radiology. Additionally, we have worked to clarify the methodological approach we intended to take with this review and the logic behind how we structured the text. I hope the revisions are to your satisfaction.

Reviewer 3 Report

Comments and Suggestions for Authors

The manuscrtpt provides a comprehensive review of the role of artificial intelligence (AI) in chest radiography across human and veterinary medicine. The comparative analysis is well-structured, and the discussion of the challenges in veterinary applications is particularly insightful. The introduction effectively sets the stage by explaining AI’s capabilities and its applications in medical imaging. However, it could benefit from a clearer statement of the review’s specific objectives to enhance focus.

The discussion on AI in human chest radiography is well-supported with relevant studies, demonstrating its diagnostic accuracy and efficiency. The inclusion of multiple AI models and their comparative performance metrics strengthens the review. However, while these studies are well-documented, some areas require deeper analysis. For instance, potential biases in AI models due to dataset limitations could be expanded upon, particularly in relation to the availability of open-source human datasets compared to the more constrained datasets in veterinary medicine.

The section on AI applications in veterinary medicine is informative but somewhat less developed than the human medicine section. While it covers key studies, it does not always provide sufficient detail regarding dataset sizes, model architectures, or validation methods. Adding more context on these aspects would improve clarity. Additionally, challenges specific to veterinary applications, such as interspecies variability, are mentioned but could benefit from a more structured discussion, possibly with proposed solutions.

The conclusion effectivly summarizes the findings and emphasizes the importance of AI as a supportive tool rather than a replacement for radiologists and clinicians. However, it would be useful to suggest specific research directions or methodologies that could help bridge the gap between human and veterinary AI applications. Addressing how lessons from human AI models can be translated into veterinary practice with greater specificity would enhance the impact of the conclusion.

The writing style is generally clear and professional, but there are instances of awkward phrasing and minor grammatical errors that should be refined for better readability. Some sentences are overly complex and could be simplified without losing their technical accuracy. A careful proofreading to improve clarity and conciseness would be beneficial. Additionally, a consistent citation style should be maintained throughout the manuscript to ensure uniformity.

Typos with line numbers

  • Line 24: "at the beginning, focusing on species" → "initially focusing on species"
  • Line 42: "enabling problem-solving across various domains [42]. At the core of AI methodologies, Machine Learning allows artificial agents" → "enabling problem-solving across various domains. At the core of AI methodologies, machine learning allows artificial agents"
  • Line 68: "remained constant (e.g., <15% for chest radiographic studies)" → "remained constant (e.g., less than 15% for chest radiographic studies)"
  • Line 81: "achieving an overall accuracy of 82.71%, with a sensitivity of 68.42% and a specificity of 87.09%" → "achieved an overall accuracy of 82.71%, with a sensitivity of 68.42% and a specificity of 87.09%"
  • Line 138: "Carloni & Tsaftaris (2025) introduced a framework to enhance AI model robustness against domain shifts in chest radiographs" → "Carloni and Tsaftaris (2025) introduced a framework to enhance AI model robustness against domain shifts in chest radiographs"
  • Line 292: "requires further study to fully assess its potential and adapt its techniques to the specificities of species such as dogs and cats" → "requires further study to fully assess its potential and adapt its techniques to the specific characteristics of species such as dogs and cats"
  • Line 307: "feature extraction by detecting anatomical landmarks." → "feature extraction through anatomical landmark detection."
  • Line 330: "Studies in human medicine tend to benefit from significantly larger datasets that are public, which allow for more robust statistical models and more accurate validation of results." → "Studies in human medicine tend to benefit from significantly larger publicly available datasets, allowing for more robust statistical models and more accurate validation of results."

Author Response

COMMENTS 1

  • Line 24: "at the beginning, focusing on species" → "initially focusing on species"
  • Line 42: "enabling problem-solving across various domains [42]. At the core of AI methodologies, Machine Learning allows artificial agents" → "enabling problem-solving across various domains. At the core of AI methodologies, machine learning allows artificial agents"
  • Line 68: "remained constant (e.g., <15% for chest radiographic studies)" → "remained constant (e.g., less than 15% for chest radiographic studies)"
  • Line 81: "achieving an overall accuracy of 82.71%, with a sensitivity of 68.42% and a specificity of 87.09%" → "achieved an overall accuracy of 82.71%, with a sensitivity of 68.42% and a specificity of 87.09%"
  • Line 138: "Carloni & Tsaftaris (2025) introduced a framework to enhance AI model robustness against domain shifts in chest radiographs" → "Carloni and Tsaftaris (2025) introduced a framework to enhance AI model robustness against domain shifts in chest radiographs"
  • Line 292: "requires further study to fully assess its potential and adapt its techniques to the specificities of species such as dogs and cats" → "requires further study to fully assess its potential and adapt its techniques to the specific characteristics of species such as dogs and cats"
  • Line 307: "feature extraction by detecting anatomical landmarks." → "feature extraction through anatomical landmark detection."
  • Line 330: "Studies in human medicine tend to benefit from significantly larger datasets that are public, which allow for more robust statistical models and more accurate validation of results." → "Studies in human medicine tend to benefit from significantly larger publicly available datasets, allowing for more robust statistical models and more accurate validation of results."

RESPONSE 1

We thank you for the various suggested corrections. We have made the necessary changes to the sections you highlighted. We hope they meet your approval.

COMMENTS 2

The discussion on AI in human chest radiography is well-supported with relevant studies, demonstrating its diagnostic accuracy and efficiency. The inclusion of multiple AI models and their comparative performance metrics strengthens the review. However, while these studies are well-documented, some areas require deeper analysis. For instance, potential biases in AI models due to dataset limitations could be expanded upon, particularly in relation to the availability of open-source human datasets compared to the more constrained datasets in veterinary medicine.

The section on AI applications in veterinary medicine is informative but somewhat less developed than the human medicine section. While it covers key studies, it does not always provide sufficient detail regarding dataset sizes, model architectures, or validation methods. Adding more context on these aspects would improve clarity. Additionally, challenges specific to veterinary applications, such as interspecies variability, are mentioned but could benefit from a more structured discussion, possibly with proposed solutions.

The conclusion effectivly summarizes the findings and emphasizes the importance of AI as a supportive tool rather than a replacement for radiologists and clinicians. However, it would be useful to suggest specific research directions or methodologies that could help bridge the gap between human and veterinary AI applications. Addressing how lessons from human AI models can be translated into veterinary practice with greater specificity would enhance the impact of the conclusion.

The writing style is generally clear and professional, but there are instances of awkward phrasing and minor grammatical errors that should be refined for better readability. Some sentences are overly complex and could be simplified without losing their technical accuracy. A careful proofreading to improve clarity and conciseness would be beneficial. Additionally, a consistent citation style should be maintained throughout the manuscript to ensure uniformity

RESPONSE 2

I sincerely appreciate your suggestions once again. Our goal was to present the most recent and significant studies on AI applications in conventional radiology, both in human and veterinary medicine. From there, we aimed to examine the current limitations and potential benefits.

We have made an effort to improve the text based on your recommendations, hoping it meets your approval.

Reviewer 4 Report

Comments and Suggestions for Authors

In the introduction, the authors rightly highlight the advantages of AI in terms of speed, accuracy, and efficiency. This is indeed accurate—human radiologists are susceptible to neurocognitive impairments caused by factors such as sleep deprivation, and increased workload can negatively impact their performance. For instance, the detectability of lesions significantly decreases when viewing time is reduced to under four seconds. A more detailed overview of this issue can be found here: https://rjlm.ro/index.php/arhiv/1022.

When discussing the use of AI in interpreting human radiographs, the authors appropriately point out the critical role of the datasets used for training. The composition of these datasets can introduce biases into the AI models (see: https://www.mdpi.com/2227-9032/9/11/1545). Particularly commendable is the authors’ mention of the importance of ensuring equitable diagnostic performance across different demographic groups without compromising overall accuracy. This is a key concern, as AI systems have been shown to potentially exhibit racial bias (https://pubmed.ncbi.nlm.nih.gov/31649194/).

Regarding the application of AI in veterinary radiology, the authors could expand on the differences between interpreting and annotating human and animal chest X-rays (CXR). While radiology is a well-established specialty in human medicine, many veterinarians may lack the same level of training in radiographic interpretation. This can affect the quality of image annotations used in AI training datasets. Providing some context on the professional backgrounds of human versus veterinary radiologists would help clarify how such disparities may impact AI development in this field. It is not uncommon for veterinarians to consult with human radiologists for more accurate diagnostics.

Additionally, the use of AI in the European Union is subject to regulation under the AI Act, which categorizes AI systems into various risk levels. It would be valuable for the authors to discuss how such regulations might influence the development and application of AI in veterinary medicine.

Overall, I commend the authors for their comprehensive review. I recommend the article for publication, ideally incorporating the points raised above for further improvement.

Author Response

COMMENTS 1

In the introduction, the authors rightly highlight the advantages of AI in terms of speed, accuracy, and efficiency. This is indeed accurate—human radiologists are susceptible to neurocognitive impairments caused by factors such as sleep deprivation, and increased workload can negatively impact their performance. For instance, the detectability of lesions significantly decreases when viewing time is reduced to under four seconds. A more detailed overview of this issue can be found here: https://rjlm.ro/index.php/arhiv/1022.

When discussing the use of AI in interpreting human radiographs, the authors appropriately point out the critical role of the datasets used for training. The composition of these datasets can introduce biases into the AI models (see: https://www.mdpi.com/2227-9032/9/11/1545). Particularly commendable is the authors’ mention of the importance of ensuring equitable diagnostic performance across different demographic groups without compromising overall accuracy. This is a key concern, as AI systems have been shown to potentially exhibit racial bias (https://pubmed.ncbi.nlm.nih.gov/31649194/).

Regarding the application of AI in veterinary radiology, the authors could expand on the differences between interpreting and annotating human and animal chest X-rays (CXR). While radiology is a well-established specialty in human medicine, many veterinarians may lack the same level of training in radiographic interpretation. This can affect the quality of image annotations used in AI training datasets. Providing some context on the professional backgrounds of human versus veterinary radiologists would help clarify how such disparities may impact AI development in this field. It is not uncommon for veterinarians to consult with human radiologists for more accurate diagnostics.

Additionally, the use of AI in the European Union is subject to regulation under the AI Act, which categorizes AI systems into various risk levels. It would be valuable for the authors to discuss how such regulations might influence the development and application of AI in veterinary medicine.

Overall, I commend the authors for their comprehensive review. I recommend the article for publication, ideally incorporating the points raised above for further improvement.

RESPONSE 1

We thank you for your valuable comments and suggestions, which have been duly taken into consideration and incorporated into the revised text. I found the article about biases and errors related to the human factor—such as workload and fatigue—very interesting. I believe this is absolutely true, even in veterinary medicine. We didn’t go into too much detail about the legislation, as our main goal was to compare the most recent articles in diagnostic imaging and see whether there could be concrete opportunities for future studies. However, we did include some ethical issues related to the use of AI in the revised text. We trust they will meet with your approval.

Reviewer 5 Report

Comments and Suggestions for Authors

General

The authors address a very interesting topic on AI for chest radiography comparing AI system for humans and animals. However, the manuscript is lacking a clear structure. Lots of paper are cited without a clear (search) strategy. Authors could try to cluster papers with the same topic or area of research. The conclusion is lacking a vision on the goal of AI in (veterinary) medicine.

Introduction:

The author should explain clearly the terminology. Deep learning is not mentioned, but all current AI systems are based on deep learning.

The author make an unrealistic representation of AI in medicine. AI has not yet transformed medicine yet. Yet still all radiological exams are read by radiologist, of which a small subset with AI. Also the AI can not operate standalone and requires a check by a human expert/radiologists. Also, the added value of AI in clinical practice has not yet been demonstrated widely. Although AI are usually fast, the use of AI in clinical practice may slow down the radiologist.

The authors should explain why AI is needed in veterinary radiology. Is it expertise? Diagnostic errors? Low on personnel? Claims?

Main body

There was no search strategy. How did the authors find these articles? Was there a selection?

The authors list a lot of papers without a clear structure. Subheadings with specific topic might be useful. What is the message?

Anatomically guided inpainting does not seem a valuable clinical application!

The structure is also lacking in the veterinary applications part.

Authors could try to put some of studies in perspective. Some of the studies mentioned are with small groups for instance.

The authors give a very optimistic vision on AI, even though performance number of some studies are not great.

Conclusion

The arguments on generalizability and how AI should be used (as assistance) are valid.

Cross learning is probably not the best way to improve vet AI. More data is needed.

Also human medicine is more mature than vet medicine. For instance, we do look at heart size in humans, but also a lot of other parameters. Therefore the heart size alone is less important.

The conclusion is lacking a vision on why AI should be used.

Also outcome for patients/animals should also be a main goal when technical innovations are introduced.

Table 1 is lacking a caption. Authors should point out that this list of human AI studies is far from complete.

Author Response

COMMENTS 1

The authors address a very interesting topic on AI for chest radiography comparing AI system for humans and animals. However, the manuscript is lacking a clear structure. Lots of paper are cited without a clear (search) strategy. Authors could try to cluster papers with the same topic or area of research. The conclusion is lacking a vision on the goal of AI in (veterinary) medicine.

Introduction:

The author should explain clearly the terminology. Deep learning is not mentioned, but all current AI systems are based on deep learning.

The author make an unrealistic representation of AI in medicine. AI has not yet transformed medicine yet. Yet still all radiological exams are read by radiologist, of which a small subset with AI. Also the AI can not operate standalone and requires a check by a human expert/radiologists. Also, the added value of AI in clinical practice has not yet been demonstrated widely. Although AI are usually fast, the use of AI in clinical practice may slow down the radiologist.

The authors should explain why AI is needed in veterinary radiology. Is it expertise? Diagnostic errors? Low on personnel? Claims?

Main body

There was no search strategy. How did the authors find these articles? Was there a selection?

The authors list a lot of papers without a clear structure. Subheadings with specific topic might be useful. What is the message?

Anatomically guided inpainting does not seem a valuable clinical application!

The structure is also lacking in the veterinary applications part.

Authors could try to put some of studies in perspective. Some of the studies mentioned are with small groups for instance.

The authors give a very optimistic vision on AI, even though performance number of some studies are not great.

Conclusion

The arguments on generalizability and how AI should be used (as assistance) are valid.

Cross learning is probably not the best way to improve vet AI. More data is needed.

Also human medicine is more mature than vet medicine. For instance, we do look at heart size in humans, but also a lot of other parameters. Therefore the heart size alone is less important.

The conclusion is lacking a vision on why AI should be used.

Also outcome for patients/animals should also be a main goal when technical innovations are introduced.

Table 1 is lacking a caption. Authors should point out that this list of human AI studies is far from complete

RESPONSE 1

Thank you very much for your suggestions. We have worked to clarify the methodology used in this review. Our aim was to compile the key studies on the use of AI in both human and veterinary fields, focusing on clinical and diagnostic applications. We have made an effort to include the most recent and significant studies. Additionally, we have aimed to outline the main limitations that hinder its everyday use and compare them with the strengths, particularly focusing on the field of veterinary medicine. We also emphasized that, at present, the use of AI is still limited, and that the doctors—whether veterinary or human—remains at the center of the work. We appreciate your comments and have worked to improve the text based on your feedback.

Round 2

Reviewer 5 Report

Comments and Suggestions for Authors

Thank you for the revision of the manuscript.

The manuscript significantly improved, and the goal of the paper is much clearer. Still there was no clear search strategy. Therefore, the paper is more a (critical) viewpoint of the authors, than a clear representation of the current research and value of AI in practice.

The introduction and conclusion are improved. The methods section still could benefit from clustered topics.

Author Response

COMMENTS 1

Thank you for the revision of the manuscript.

The manuscript significantly improved, and the goal of the paper is much clearer. Still there was no clear search strategy. Therefore, the paper is more a (critical) viewpoint of the authors, than a clear representation of the current research and value of AI in practice.

The introduction and conclusion are improved. The methods section still could benefit from clustered topics.

RESPONSE 1

Thank you for your comment. We have tried to revise the final part of the introduction by clarifying our objective and the methodology we followed in the preparation of this review on AI, also aiming to suggest possible ideas for future studies in veterinary medicine. We hope this revision meets your expectations, and we thank you once again.
